# Charge redistribution of a spatially differentiated ferroelectric Bi$_4$Ti$_3$O$_{12}$ single crystal for photocatalytic overall water splitting

Guangri Jia [1,2,6], Fusai Sun[3,6], Tao Zhou[2], Ying Wang [4], Xiaoqiang Cui [5], Zhengxiao Guo [2] ✉, Fengtao Fan [3] ✉ & Jimmy C. Yu [1] ✉

Artificial photosynthesis is a promising approach to produce clean fuels via renewable solar energy. However, it is practically constrained by two issues of slow photogenerated carrier migration and rapid electron/hole recombination. It is also a challenge to achieve a 2:1 ratio of H$_2$ and O$_2$ for overall water splitting. Here we report a rational design of spatially differentiated two-dimensional Bi$_4$Ti$_3$O$_{12}$ nanosheets to enhance overall water splitting. Such a spatially differentiated structure overcomes the limitation of charge transfer across different crystal planes in a single crystal semiconductor. The experimental results show a redistribution of charge within a crystal plane. The resulting photocatalyst produces 40.3 μmol h$^{-1}$ of hydrogen and 20.1 μmol h$^{-1}$ of oxygen at a near stoichiometric ratio of 2:1 and a solar-to-hydrogen efficiency of 0.1% under simulated solar light.

Photocatalytic water splitting is regarded as one of the promising and cost-effective technologies to mitigate energy and environmental challenges[1-4]. For further practical applications and large-scale production, it is necessary to realize effective overall water splitting (OWS) into hydrogen and oxygen. However, for most photocatalytic systems, achieving a stoichiometric ratio of H$_2$:O$_2$ = 2:1 is very challenging[5-7]. One of the reasons is the rapid recombination of photogenerated carriers, especially when there are detrimental defects and complex grain boundaries in the photocatalyst[8-11]. Compared with integrated semiconductors or redox media (I$^-$/IO$_3^-$ and Fe$^{2+}$/Fe$^{3+}$ as electron mediators), direct OWS into H$_2$ and O$_2$ via a one-step excitation on particulate photocatalysts would be more economical, particularly for large-scale hydrogen production[12-16]. Unfortunately, single-component photocatalytic system is more prone to photogenerated

carrier recombination. Thus, it is important to resolve such issues by techniques such as catalyst modification.

In the past few decades, many photocatalysts have been discovered for photocatalytic hydrogen evolution[17-23]. However, those capable of OWS by one-excitation are rare[24-26]. As a typical example, Domen's group reported that a series of SrTiO$_3$ materials could be used for photocatalytic OWS to produce hydrogen and oxygen[1,27]. Tantalate- and niobate-based compounds also exhibit excellent OWS activity when modified with metal ions[28-32]. However, those usually suffer from a large band gap and rapid photogenerated carrier recombination, leading to a low solar-to-hydrogen efficiency of less than 0.1%. Although (oxy)nitrides, (oxy)sulfides, and oxy-halides with excellent light absorption properties show photocatalytic OWS activity, poor stability limits their practical consideration[9,33-35]. Thus, it is

[1]Department of Chemistry, The Chinese University of Hong Kong, Shatin, New Territories, Hong Kong 999077, China. [2]Department of Chemistry, The University of Hong Kong, Pokfulam Road, Hong Kong 999077, China. [3]State Key Laboratory of Catalysis, Dalian National Laboratory for Clean Energy, iChEM, Dalian Institute of Chemical Physics, Chinese Academy of Sciences, Dalian 116023, China. [4]Department of Applied Biology and Chemical Technology, The Hong Kong Polytechnic University, Hung Hom, Kowloon, Hong Kong SAR, China. [5]State Key Laboratory of Automotive Simulation and Control, School of Materials Science and Engineering, Key Laboratory of Automobile Materials of MOE, Jilin University, Changchun 130012, China. [6]These authors contributed equally: Guangri Jia, Fusai Sun. ✉e-mail: zxguo@hku.hk; ftfan@dicp.ac.cn; jimyu@cuhk.edu.hk

pivotal to develop efficient particulate photocatalysts that are highly active and stable for OWS.

$Bi_4Ti_3O_{12}$ (BTO) is a structural two-dimensional layered ferroelectric material that consists of the regular stacking of bismuth oxygen $[Bi_2O_2]^{2+}$ slabs and perovskite-like $[Bi_2Ti_3O_{10}]^{2-}$ units. Such a sandwiched structure may facilitate the separation of photogenerated electron–hole pairs by means of the internal electric field formed between the interlayers, especially the introduction of external factors to enhance this effect[36–39]. Due to its appropriate band structure, BTO possesses the advantages of desirable visible light absorption and conduction band[40]. However, without a specific crystal plane to separate the electron–hole pairs, pure BTO crystalline particles are not effective for OWS, because the photogenerated carriers are prone to recombine in the bulk phase. This problem can be partially overcome by the introduction of appropriate heterojunctions for OWS[41,42]. However, such heterogeneous systems often show complex carrier dynamics and possible unpleasant surface recombination centers due to mismatches of lattice parameters and/or the local defect environment. It is highly desirable if the OWS can be realized in a single-component system. Moreover, reconstruction of a spatial structure that can re-distribute the charges would be needed for developing practical applications.

With due consideration of the above, we designed BTO single crystals with a spatially differentiated structure (SDS) created by a simple top-down hydrothermal method. This SDS can induce effective charge separation/redistribution, as summarized below. The bulk BTO acts as a starting metal oxide and can be specifically etched by hydrohalic acid. Hydrohalic acids (except HF) can preferentially etch the bismuth oxygen $[Bi_2O_2]^{2+}$ layer of BTO, forming gradually hollowed or spatially differentiated open ends of the BTO nanosheets over time without the requirement for an additional template or any special techniques[43–45]. At the same time, the presence of a certain concentration of acid is necessary to prevent the nucleation of undesirable bismuth oxyhalide and $TiO_2$. The specific etching does not alter the original material structure and maintains the single crystals free from grain boundaries and defects over the period of treatment, which will ensure maximum utilization of photogenerated electrons and holes without photocorrosion. Meanwhile, light-induced spatial charge redistribution was achieved, as visualized by advanced nanometer-resolution surface photovoltage microscopy (SPVM) in combination with spatially resolved surface photovoltage spectroscopy (SRSPS) using kelvin probe force microscopy (KPFM). To achieve the stoichiometric ratio of $H_2:O_2 = 2:1$ and efficient performance, suitable co-catalysts, such as Rh-CrO$_x$ and CoO$_x$ were deposited on the photocatalyst in favor of the forward reaction direction to split water into hydrogen and oxygen. Rh-CrO$_x$ promotes hydrogen production and prevents the reverse reaction and side reaction, and CoO$_x$ catalyzes water oxidation to oxygen. We demonstrate that the BTO SDS is capable of evolving hydrogen and oxygen for one-step-excitation OWS under simulated solar light at a solar-to-hydrogen efficiency of 0.1%, 212 times greater than pristine single-crystal BTO.

## Results and discussion
### Synthesis and characterization of BTO photocatalysts
As a layered structural two-dimensional material, BTO is characterized by regular stacking of bismuth oxygen $[Bi_2O_2]^{2+}$ slabs and perovskite-like $[Bi_2Ti_3O_{10}]^{2-}$ blocks, in which Bi–O bond can be etched via the interaction of proton and halide ion. Briefly, pristine single-crystal BTO was firstly prepared by a salt-assisted solid-state reaction. Due to the different structural stability of exposed crystal facets, the $[Bi_2O_2]^{2+}$ structure of {001} crystal facet and Bi–O bond of {010} crystal facet of BTO can be selectively preferentially etched (Supplementary Fig. 1 and Supplementary Note 1). Also, the Ti–O bond of BTO {001} will prevent further etching of {001} crystal facet after etching $[Bi_2O_2]^{2+}$, ultimately leading to the formation of a hollow structure along the edge region.

Scanning electron microscopy (SEM) images show the pristine BTO with an evident nanosheet morphology. When HCl etching was conducted, the etched edge gradually became highly visible with the extension of the etching time to 2 h (Fig. 1a and Supplementary Fig. 2). However, the particle size or crystallite size of samples had no significant change, but the obvious difference of specific surface area occurs before and after acid etching (Supplementary Fig. 3 and Supplementary Table 1). When the etching time extended to 4 h, the BTO-4 shows that the nanosheets are broken with impurities due to nucleation of $TiO_2$ (Supplementary Fig. 4). X-ray diffraction (XRD) pattern indicates the same crystal structure between the pristine and the etched BTO within 2 h (Fig. 1b). However, the crystallinity of the etched BTO is gradually weakened with further extension of etching time (Fig. 1c). Unsurprisingly, the over-etched BTO-4 shows a mixed structure of BTO and $TiO_2$ (Supplementary Figs. 5 to 6 and Supplementary Note 2). Similarly, under the same concentration as HCl, the HBr-treated BTO sample shows similar crystal structure and morphology (Supplementary Figs. 7 and 8). The control experiments were performed with $HNO_3$ and $H_2SO_4$ treated pristine BTO under the same concentration as HCl. The XRD results indicate the same crystal structure as BTO (Supplementary Fig. 9). And the edge region of BTO has not been etched (Supplementary Fig. 10). Further, an atomic force microscope (AFM) was used to investigate the thickness change before and after etching for different periods. No significant change appears in the overall thickness before and after etching and between the center and edge of the etched BTO (Supplementary Fig. 11). Furthermore, a highly hollow edge structure was observed by the SEM image, and the corresponding elemental mapping reveals the uniform distributions of Bi and Ti elements (Supplementary Fig. 12). Other SEM images, the corresponding elemental mapping, and linear scan elemental analysis are provided in Supplementary Figs. 13–15, and the results proved the uniform distributions of Bi, Ti, and O elements in both the center and the edge region of BTO with and without HCl treatment.

Further characterization of the local structure of the etched BTO was carried out under high-resolution transmission electron microscopy (HRTEM). An evident thickness variation contrast occurred between the central and the edge regions of transmission electron microscopy (TEM) image because of the etching effect (Fig. 2a). The HRTEM image exhibits the same lattice orientation and lattice fringes between the edge and the central regions of the etched BTO (Figs. 2b,c), indicating the single crystalline nature of the structure. Furthermore, the same diffraction patterns further confirm the single-crystal structure by selected area electron diffraction (SAED) at different locations (Figs. 2d and 2e). Accordingly, the chemical valence states of elements were also characterized by X-ray photoelectron spectroscopy (XPS). Due to the gradually etched depth of the structure, the chemical states of the surface elements change in a dynamically evolving manner, especially for Ti and Bi. Evidently, as shown in Fig. 2f, the binding energy of Ti 2$p$ gradually shifts towards a higher value, which comes from the increasing proportion of the Ti–O structures exposed on the surface because of the Ti element binding more high-energy O (hydroxyls and water molecules). Due to the change of lattice oxygen (Bi–O and Ti–O) structure on the exposed surface, the binding energy of O 1$s$ shifts to higher value at the same time (Supplementary Fig. 16a). In contrast, the corresponding Bi 4$f$ peak slightly shifts downward in binding energy with increasing extent of the acid etching, because the outside high binding energy Bi of $[Bi_2O_2]^{2+}$ layer and $[Bi_2Ti_3O_{10}]^{2-}$ was etched away. Although the binding energies of Bi and Ti evolve, the main structure of samples keeps unchanged during the etching process (Fig. 2g). No Cl element was observed during the etching process, indicating that the nucleation of bismuth oxyhalide was inhibited (Supplementary Fig. 16b). In addition, the structural change should be accompanied by the evolution of the band structure. The band gap slightly increases with the extension of

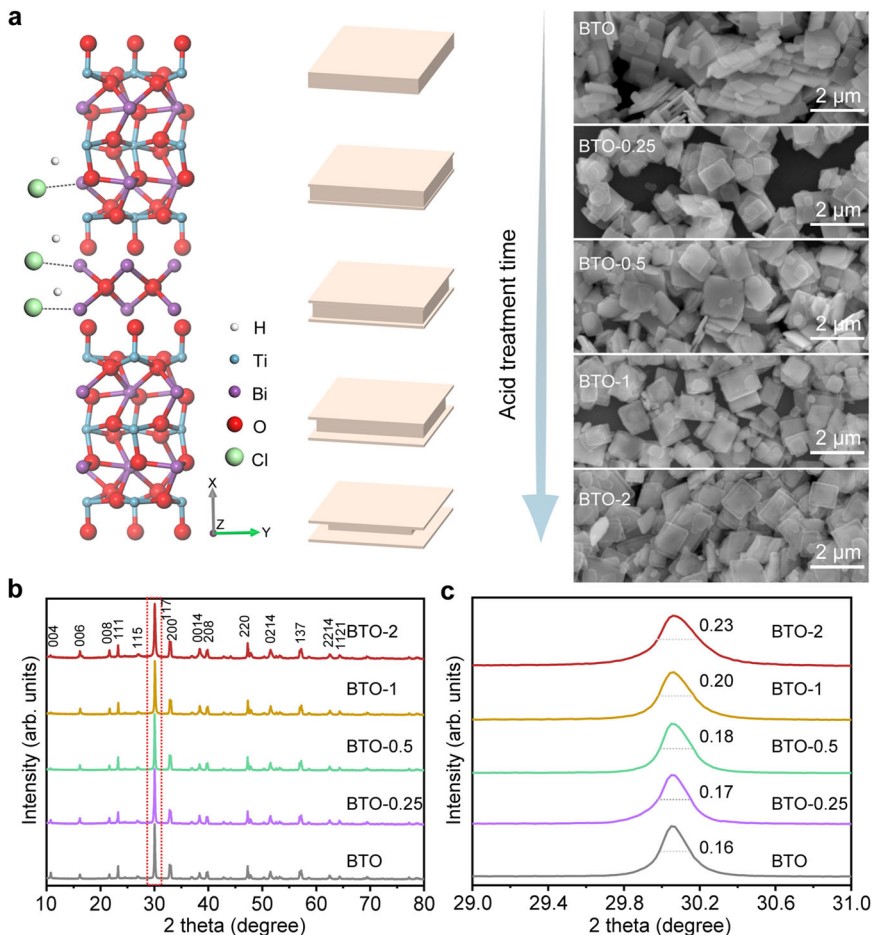

**Fig. 1 | Morphology and crystal structure analysis of photocatalysts. a** Crystal structure and the morphology evolution process of BTO. **b** XRD patterns for as-prepared BTO at different HCl-treated durations. **c** Enlarged XRD patterns in (**b**).

the etching, which is due to the change of the band structure by the spatial size effect, which is mainly from the ultrathin hollow edge (Supplementary Figs. 17–19). With the increase of edge etching depth, the trend of band structure change is more obvious. Meanwhile, the difference in the band structures between the unetched part and adjacent etched BTO will form "homojunction", which are beneficial to the transfer of electrons to the surface of the photocatalyst. Also, due to $TiO_2$ impurities resulting from structural destruction caused by the excessive etching, the BTO-4 further shows the bigger band gap and evident differences from BTO in light absorption behavior. However, such a small change of bandgaps is not expected to influence the photocatalytic performance.

## Charge transfer dynamic and lifetime

Further understanding the charge distribution on the crystal surface is helpful to explore the process of photocatalytic water splitting. SPVM is an effective tool for spatial correlation between photogenerated charge distribution and structural morphology, as well as the dependence between light absorption and carrier migration and transfer[46,47]. As shown in Figs. 3a–f, photogenerated electrons were imaged in the surface due to negative SPV. With the increasing wavelength of illumination light, the intensity of the SPV decreases, which is consistent with the behavior of the light absorption spectrum, which proves that the charge generation is driven by light (Supplementary Fig. 17 and Fig. 3g). The difference is that when the light irradiated the surface of photocatalyst, the charge distribution intensity varies between the edge and the middle of the sample. This phenomenon is very interesting and differs from most of the results reported in the

literatures[48–51]. We speculate the carrier distribution is dependent on the space geometry. Differential SPV analysis was performed at different positions of a given facet and the differences were compared at the three positions. The corresponding results are shown in Figs. 3a–f. The SPV is significantly different with the change of light wavelength at the three locations of α1, α2 and α3. Therefore, the intensity of SPV varies from the edge to the middle position, which induces a directional movement of surface-generated electrons, leading to effective photogenerated carrier separation compared with pristine BTO (Supplementary Fig. 20). Combined with the previous results, we find that the electron transfer is related to the spatial morphology and structure. The hollow regions with etched edges possess abundant photogenerated electrons, which is mainly because the regions with thin edges can accelerate electron migration to the surface from the bulk and center. In order to further determine the transfer and the regional distribution of electrons, the deposition of Au on BTO was performed with and without acid treatment. The random distribution of Au nanoparticle can be observed on the surfaces of pristine BTO in Fig. 3h. Not surprisingly, it is found to be preferentially deposited on the edge area of surface crystal facets of BTO-2 in Fig. 3i, which shows a special behavior, that is, spatially differentiated structure induces the differentiated distribution of electrons. A fast electron transfer dynamic will be promoted due to the electron differentiated distribution, which is beneficial to photogenerated charge separation for efficient OWS (Fig. 3j).

The process of photocatalysis is largely limited by the separation and transfer of photogenerated charge carriers[52,53]. The lifetime and transfer dynamic were characterized by photoluminescence (PL) and

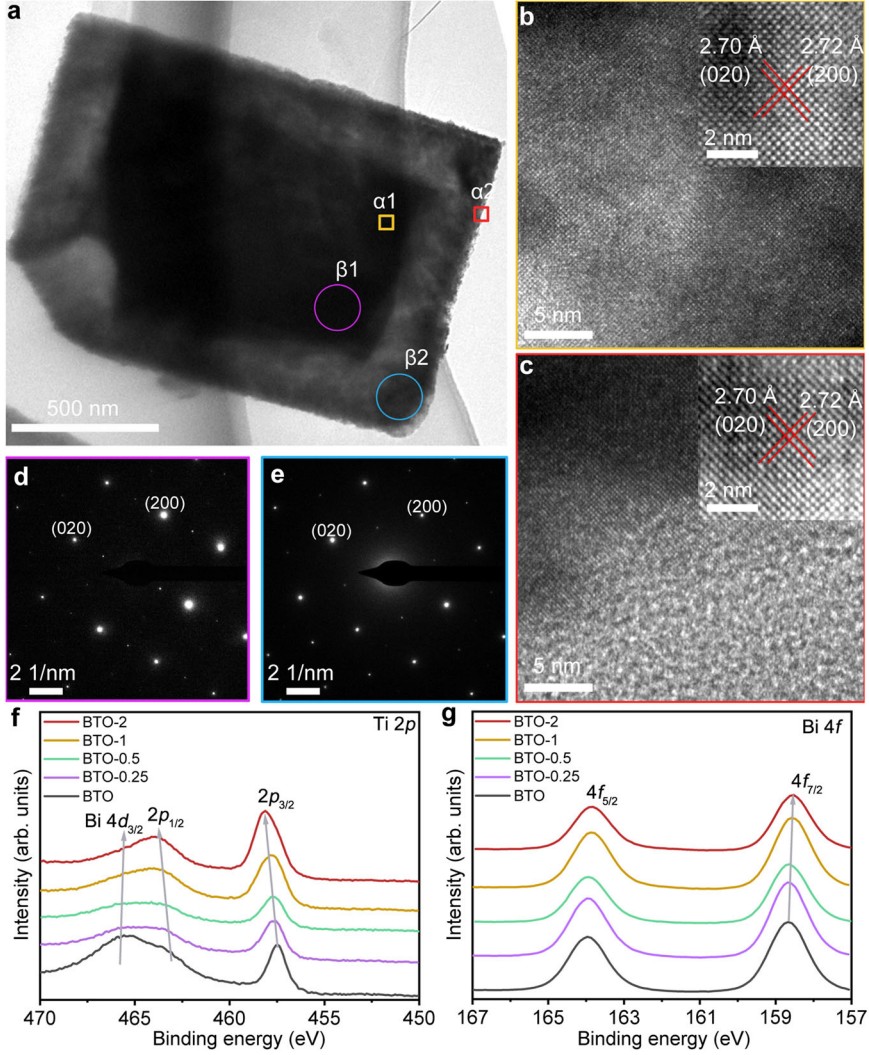

**Fig. 2 | Surface structure characterization of photocatalysts. a** TEM image of BTO-2. **b** HRTEM image of yellow frame in α1 area of (**a**). **c** HRTEM image of red frame in α2 area of (**a**). **d** SAED pattern of purple circle in β1 area of (**a**). **e** SAED pattern of blue circle in β1 area of (**a**). **f** High-resolution XPS spectra of Ti 2*p*. **g** High-resolution XPS spectra of Bi 4*f*.

time-resolved photoluminescence (TRPL). Both pristine BTO and etched BTO show an evident PL peak at 485 nm (Supplementary Fig. 21). The PL intensity of the etched BTO gradually decreases with etching time, indicating a decrease in the number of excitons due to carries' effective separation. We carried out TRPL to monitor the charge carrier dynamics on the nanosecond time-scale. As shown in the Fig. 4a, the carrier lifetime curve conforms to a multi-exponential decay function, which can be fitted by different parameters. The average lifetime is 3.64 ns for pristine BTO, which is longer than that of the etched BTO. And the average carrier lifetime gradually reduces with the duration of HCl treatment (3.28 ns for BTO-0.25, 2.71 ns for BTO-0.5, 2.34 ns for BTO-1, and 1.69 ns for BTO-2), representing that charge carriers' transfer and lifetime is dependent on spatial morphology. Specifically, the faster decay lifetime of $\tau_1$ of etched BTO is longer than that of pristine BTO and the contribution of fast decay of $\tau_1$ increases largely from 34% for BTO to 85% for BTO-2, confirming the suppression of the inter-band recombination and fast electron transfer of the etched BTO due to the homojunction between the edge and the center. Also, combined with the morphological analysis, the decreased lifetime is gentle at the initial stages of etching, which is because the etching time is relatively short and only the surface Bi−O layer is etched (Fig. 4b). With increasing duration of the etching, the carrier's lifetime

of the sample rapidly declines, which is positively correlated to the etching depth due to the formation of the homojunction and the ultrathin edge, where the electrons of the etched edge will be accelerated and transferred to the surface compared with the center part of the BTO (Fig. 4c). Moreover, we further demonstrated the rapid interfacial electron transport performance. The etched BTO shows a larger photocurrent than that of the pristine BTO. And it is found that a positively correlation of the photocurrent with the etching depth for etched BTO (Supplementary Fig. 22a). The electrochemical impedance spectroscopy (EIS) measurements and analyses also reveal fast electron transport of the etched BTO. The radius of the fitted curve reflects the strength of the transfer of semiconductor electrons, that is, the smaller the radius indicates the faster the electron transfer[54,55]. As shown in the impedance curves (Supplementary Fig. 22b), the BTO-2 shows the smallest electrochemical impedance, implying the fastest electron transport among the photocatalysts.

## Photocatalytic performance

As the spatial charge distribution caused by the etching effect changes, the photocatalytic OWS will be significantly different. We carried out the photocatalytic OWS experiment on the obtained samples under ultraviolet−visible (UV−Vis) light simulated by a xenon lamp. First, the

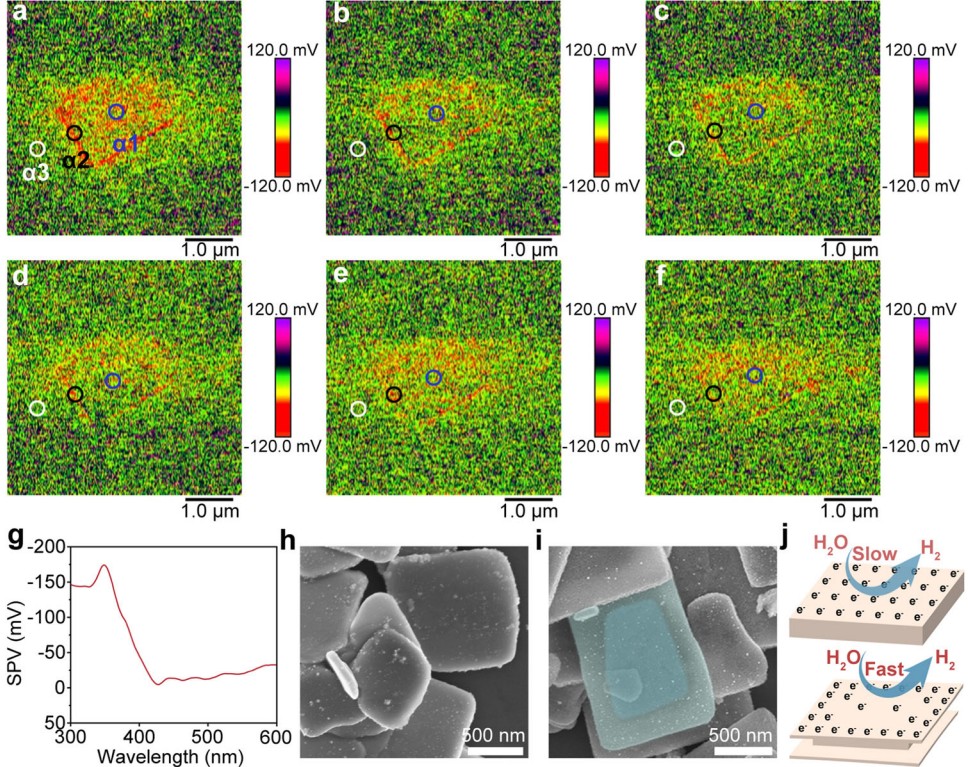

**Fig. 3 | Spatial distribution of charge separation. a–f** SPVM images at different wavelengths of 350 nm, 360 nm, 370 nm, 380 nm, 400 nm, and 420 nm. Blue circle (α1), black circle (α2) and white circle (α3) represent the center region of BTO-2, the edge region of BTO-2, and the sapphire substrate, respectively. **g** SRSPS of BTO-2 nanoparticle as a function of photoexcited wavelength under illumination. SEM images of Au-deposited on BTO (**h**) and BTO-2 (**i**). Dark middle region shows a lower density of photodeposited Au, while the lighter edge region shows a higher density of photodeposited Au of (**i**). **j** Schematic illustration of the distribution of photo-generated electron belonging to charge separation.

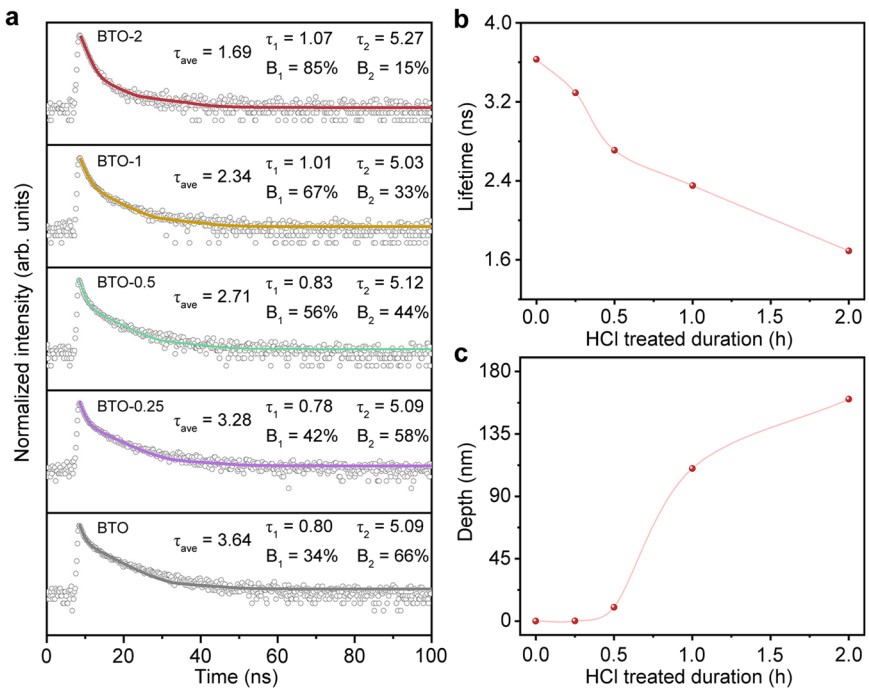

**Fig. 4 | Charge lifetime of photocatalysts. a** TRPL spectra of BTO in different etching time evolution processes, from top to bottom: BTO-2, BTO-1, BTO-0.5, BTO-0.25, and BTO. The empty dots represent the decay curves and the straight lines are the corresponding fit curves. **b** Lifetime evolution with the extended etching time. **c** Etched depth evolution with the extended etching time.

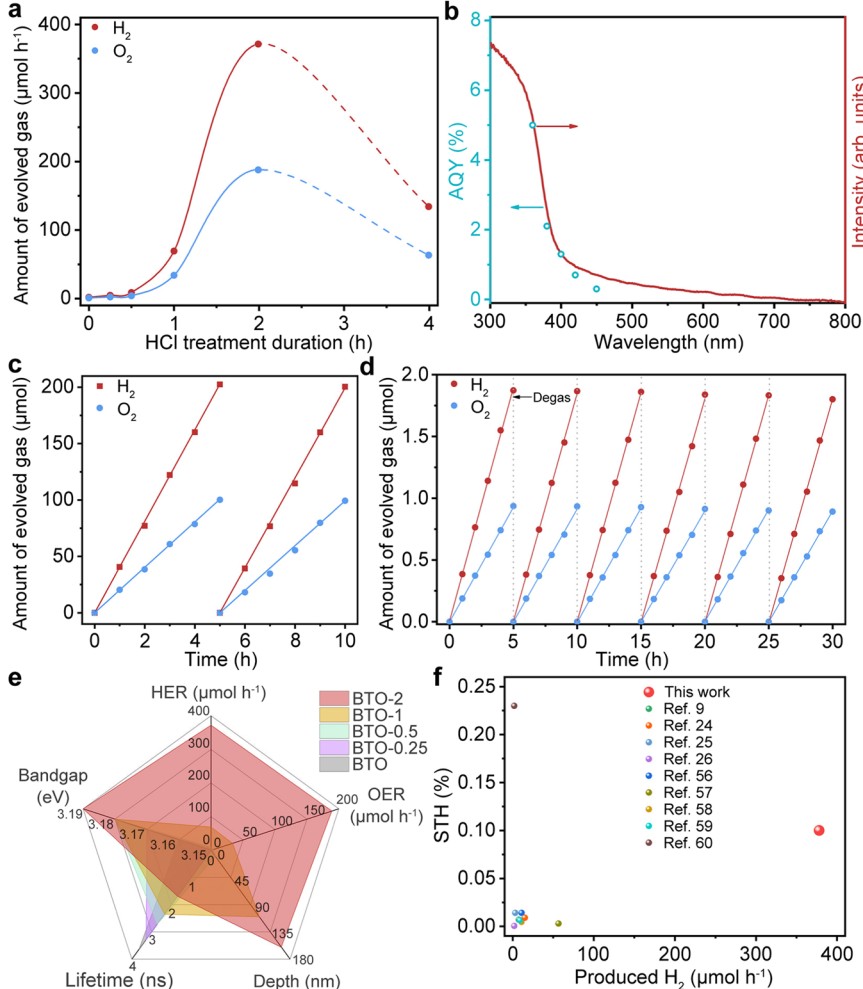

**Fig. 5 | OWS performance and AQE of photocatalysts. a** Photocatalytic OWS of BTO in different etching time evolution processes under UV–Vis light using 300 W xenon lamp. **b** AQE of BTO-2 wavelength dependence of photocatalytic activity. Conditions: 0.12 g catalyst, 0.5 wt% Rh, 100 mL deionized water, and 300 W xenon lamp. **c** Time course of OWS of BTO-2 under simulated sunlight (AM 1.5 G). The solid line is linear fitting for $H_2$ ($R^2 \geq 0.999$) and $O_2$ evolution ($R^2 \geq 0.999$). **d** Stability of photocatalyst of BTO-2 with the same condition as (**a**). The solid line is linear fitting for $H_2$ ($R^2 \geq 0.999$) and $O_2$ evolution ($R^2 \geq 0.999$). **e** Comparison of structure-activity relationship of photocatalytic water decomposition properties. **f** Comparison of the STH and produced hydrogen under xenon lamp over the BTO photocatalyst with those of reported one-excitation photocatalysts.

hydrogen evolution cocatalyst $Ru/CrO_x$ and oxygen evolution cocatalyst $CoO_x$ were deposited on the photocatalytic surface under light irradiation respectively. Figure 5a indicates the photocatalytic OWS activity of the BTO as a function of etching time, while the corresponding gas evolution rate was obtained (Supplementary Fig. 23). The photocatalytic activity increased significantly with the etching time and reached a maximum value of 371.8 µmol $h^{-1}$ ($H_2$) and 188.8 µmol $h^{-1}$ ($O_2$) for BTO-2, 212 times greater than pristine single-crystal BTO. And the ratio of hydrogen to oxygen remained the desired stoichiometric ratio throughout the process. The subsequent sharp decline of activity is due to the collapse of the structure and the presence of impurities for excessive etched BTO-4. Then, we investigated and optimized the photocatalytic activity of hydrogen evolution and oxygen evolution under different loading contents of cocatalyst due to the correlation between photocatalytic activity and cocatalyst (Supplementary Fig. 24). These results provide further evidence that photocatalytic activity is dependent on spatial morphology. Apparent quantum efficiency (AQE) is closely related to the wavelength of the radiated light as shown in Fig. 5b. The AQE value of OWS reaches 5.0% at 360 nm, 2.1% at 380 nm, and 1.3% at 400 nm, respectively. At the same time, we carried out the photocatalytic OWS at simulated sunlight (Fig. 5c). The photocatalytic activity achieved an excellent

performance, achieving 40.3 µmol $h^{-1}$ of hydrogen and 20.1 µmol $h^{-1}$ of oxygen at a near stoichiometric mole ratio of 2:1. BTO-2 was subjected to six OWS cycles for 30 h after degassing the evolved gas in each cycle, and there was little decay evolution of $H_2$ and $O_2$, demonstrating the stable photocatalytic performance during OWS under light irradiation (Fig. 5d). Finally, the solar-to hydrogen (STH) energy conversion efficiency of BTO-2 was estimated to be 0.1%. In conclusion, the spatial structural design of the single-crystal BTO photocatalyst provides a good strategy to improve water splitting, which is beneficial to light absorption and carrier separation of the photocatalyst. Also, the larger specific surface area of BTO-2 increases the dispersion of the cocatalyst, which is favorable for the improvement of photocatalytic performance. The performance change is enabled by the varying band structure, but the band gap effect is not evident during the evolution of the photocatalytic properties in this photocatalytic system. The crucial roles of lifetime and transfer of the photogenerated carriers are the main factors dominating the OWS, and are closely associated with the spatial structure change (Fig. 5e). Comparing the photocatalytic water splitting properties of the photocatalyst with those reported in the literature, the OWS performance shows excellent photoconversion efficiency among many photocatalysts (Fig. 5f and Supplementary Table 2)[9,24–26,56–60].

In summary, a spatially differentiated structure of single-crystal bismuth titanate has been synthesized by a one-step acid etching approach. Hydrochloric acid preferentially etched the middle of the edge region of the layered BTO, forming a gradually hollowed or spatially differentiated edge structure. The relative etched area can be controlled etching condition, e.g., etching time, to regulate the spatial charge distribution of the sample. The etched BTO single crystals enable the rapid migration of photoexcited charges from the center to its surface and edge regions of the nanostructure, leading to effective charge transfer and separation. The one-step-excitation photocatalytic OWS by the SDS photocatalyst results in a total solar-to-hydrogen efficiency of 0.1%, about 212 times greater than the pristine single-crystal BTO under simulated solar light. Such a strategy can be applied to design efficient and stable single-crystal photocatalysts towards practical solar energy conversion.

## Methods

### Synthesis of etched BTO nanosheets

Pristine BTO nanosheets were prepared by a conventional solid-state reaction method. A certain weight of $Bi_2O_3$ (Aladdin, USA) and $TiO_2$ (anatase, Aladdin, USA) was added in a mixture of KCl (Aladdin, USA) and NaCl (Aladdin, USA) with a mole ratio of Bi:Ti:K:Na of 4:3:160:160. The mixture was thoroughly ground for 15 minutes in a mortar and annealed at 800 °C for 12 h in static air with a ramping rate of 5 °C·min$^{-1}$. Then, the sample was washed with deionized water and ethanol to remove any KCl and NaCl for three times respectively. The obtained powder was dried at 60 °C for 8 h. Subsequently, the as-prepared BTO powder was subjected to etched process. 0.1 g of BTO was added into 20 mL aqueous solution (0.6 M HCl) under ultrasound. The mixture was transferred into in a 50 mL Teflon-lined stainless-steel autoclave at 100 °C for duration ranging from 0.25 to 4 h.

### Photodeposited Au nanoparticles on BTO photocatalysts

20 mg of samples were dispersed in 50 mL aqueous solution containing 0.8 mg mL$^{-1}$ Au$^{3+}$ (HAuCl$_4$, Aladdin, USA) without any adjustment at 5 °C by circulating cooling water. The dissolved air is completely removed in suspension for 30 min by a vacuum process. Then, the mixture was irradiated under UV–Vis light and continuously stirred for 3 h.

### Photodeposited cocatalyst on BTO photocatalysts

The Rh-Cr$_2$O$_3$ and CoO$_x$ cocatalyst was loaded on the BTO photocatalyst by a three-step photodeposition process. Aqueous solutions of 1 mg mL$^{-1}$ of Rh$^{3+}$ from RhCl$_3$·3H$_2$O (Sigma-Aldrich, USA), 1 mg mL$^{-1}$ of Cr$^{6+}$ from Na$_2$CrO$_4$ (Aladdin, USA) and 1 mg mL$^{-1}$ Co$^{2+}$ from Co(NO$_3$)$_2$·6H$_2$O (Sigma-Aldrich, USA.) were prepared. 40 mg of samples were dispersed in 100 mL of deionized water containing the desired amount of metal precursor without any adjustment at 15 °C by circulating cooling water. The dissolved air is completely removed in suspension by blowing Ar for 30 min. And then the mixture was added with Rh$^{3+}$ solution and continuously stirred under UV–Vis light for 10 min. Subsequently, the Na$_2$CrO$_4$ aqueous solution was added to the suspension with additional irradiation for 5 min. And the mass ratio between the Cr:Rh was kept a constant at 1:1. After 5 min, the Co(NO$_3$)$_2$ aqueous solution was added and irradiated for 10 min. Finally, the photocatalyst was removed, centrifuged, and washed in deionized water. The obtained powder was dried in vacuum at 60 °C for further photocatalytic testing.

### Characterizations

The powder XRD was evaluated by X-ray diffractometer (Bruker, Germany) using a Cu-Kα source (0.15418 nm). SEM images were obtained using Phenom Pro Desktop system. The TEM, HRTEM, and SAED were obtained using a JEM-2000EX system. XPS was obtained using a Thermo ESCALAB-250 instrument (USA). A micro-Raman spectrometer was conducted to obtained Raman spectra using a 532 nm laser as excitation source. UV–Vis absorption spectra were conducted using a spectrometer (Cary Series, Agilent Technologies). PL and TRPL spectra were obtained using FP-6500 fluorescence spectrometer (Jasco) and Edinburgh FLS1000 spectrophotometer with excitation wavelengths of 375 nm respectively. KPFM measurements were implemented based on a Bruker Dimension V SPM system. Brunauer-Emmett-Teller surface area was obtained by N$_2$ adsorption isotherm data using Micromeritics ASAP 2460. The equipped AFM tip was equipped with a Pt/Ir coating, with 75 kHz resonance frequency, purchased from Bruker. KPFM was based on amplitude-modulation kelvin probe force microscopy (AM-KPFM) and simultaneously provides the topography and surface potential of sample. Surface potential is the difference in surface work function between the tip and the sample. Images were acquired at scan rates of 0.5 Hz, and the tip lift height was 50 nm at tapping mode for potential mapping. The contact potential difference (CPD) is defined as the difference between the work function of the tip and the sample. The SPV image is the CPD change after light irradiation, which can be calculated as SPV = ΔCPD = CPD$_{light}$ − CPD$_{dark}$, where CPD$_{dark}$ and CPD$_{light}$ are the CPD measured in the dark and in light.

### OWS reaction

Photocatalytic reactions were carried out in an overhead-irradiation-type reactor with a closed gas circulation system. A specific amount photocatalyst loaded with cocatalyst was dispersed in 100 mL of deionized water without any adjustment. Prior to irradiate the reactor with a 300 W xenon lamp (PLS-SME300E, Beijing Perfectlight, China) or a solar simulator (AM 1.5 G, 100 mW cm$^{-2}$), the air in suspension was removed by blowing Ar. The reaction system is maintained at 288 K by a cooling water system. The evolved gas products were analyzed using gas chromatography (GC 7900 Techcomp, thermal conductivity detector, and argon as the carrier gas) equipped with a molecular sieve 5 Å column.

### STH efficiency

The STH was given according to the following equation:

$$STH(\%) = \frac{R(H_2) \times \Delta G}{I \times S} \times 100 \qquad (1)$$

where $R(H_2)$, $\Delta G$, $I$, and $S$ represent the hydrogen evolution rate during OWS, the Gibbs energy of water splitting, the energy flux of light irradiation, and the irradiated area, respectively.

### AQE measurement

The AQE was calculated according to the following equation:

$$AQE(\%) = \frac{2 \times N(H_2)}{P \times S/\lambda} \times 100 \qquad (2)$$

where $N(H_2)$, $P$, $S$, and $\lambda$ donate the number of H$_2$ molecules generated, average intensity of light irradiating liquid surface, and the irradiated area during a specific time with various band-pass filters, respectively.

### Photoelectrochemical measurement

Photocurrent-time and EIS measurements were carried out using CHI650D electrochemical workstation with a 0.5 M Na$_2$SO$_4$ (Aladdin, USA) electrolyte and 300 W Xe lamp irradiation. In this system, three-electrode setup consisting of a platinum foil, Ag/AgCl reference electrode, and FTO coated with samples counter electrode as counter

electrode, reference electrode, and working electrode, respectively, were used. EIS was recorded with a frequency range of $10^5$ to $10^{-1}$ Hz and an amplitude of 10 mV.

## DFT calculations

The DFT calculations were conducted using the Vienna ab initio simulations package (VASP)[61]. The generalized gradient approximation (GGA) within Perdew-Burke-Ernzerhof (PBE) functional was employed to describe the exchange-correlation interactions[62]. The cut-off energy was 500 eV and a Monkhost-Pack k-points of $3 \times 3 \times 1$ was applied during calculations. The energy and force convergence criterion were set to $10^{-5}$ eV and $2 \times 10^{-2}$ eV/Å respectively. The DFT-D3 method was implemented to correct Van der Waals interaction. The surface structures of BTO with seven atomic layers were constructed. The top two layers were fully relaxed and the bottom five layers were fixed. The vacuum layer was 15 Å to avoid slab interactions in building the unit cell for simulations.

The Gibbs free energy was calculated by:

$$\Delta G = \Delta E + \Delta EZPE - T\Delta S \qquad (3)$$

where $\Delta E$ is electronic adsorption energy, $\Delta EZPE$ is the zero points energy and $\Delta S$ is the entropy change. T is the temperature.

## Data availability

All the data supporting the findings of this study are available from the corresponding authors upon request.

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

## Acknowledgements

This work was financially supported by the Research Grants Council of Hong Kong (Project 14304019), the Hong Kong UGC-TRS (T23-713/22-R) and National Natural Science Foundation of China (22325205). We acknowledge the State Key Laboratory of Catalysis, Dalian National Laboratory for Clean Energy, iChEM, Dalian Institute of Chemical Physics, Chinese Academy of Sciences (spatially resolved surface photovoltage spectroscopy).

## Author contributions

G.J. and F.S. contributed equally to this work. J.C.Y. directed the project. J.C.Y., F.F., and Z.G. supervised the project. G.J. performed the synthesis, most of the characterizations, and performance tests. Y.W. and X.C. analyzed the experiment results. T.Z. carried out the DFT calculations. F.S. conducted the KPFM analysis. G.J. wrote the manuscript. All authors discussed the results and commented on the paper.

## Competing interests

The authors declare no conflicts of interest.
