## [Peer Review File · Nature Communications]

REVIEWER COMMENTS

Reviewer #1 (Remarks to the Author):

Guangri Jia et al demonstrated a single-component BTO system, distinguished by a spatially differentiated structure, which exhibits exceptional efficacy in overall water splitting. The SPVM and photocatalytic performances of this system are noteworthy. However, while the manuscript underscores the significance of this advancement in the field of particulate photocatalysts, a more detailed, step-by-step clarification is required. Additionally, certain results necessitate further verification and explicit correlation with the stated significance. Therefore, after the comprehensive major revision, I would recommend this manuscript for further publication. Specific comments are as follows:

1. The chemical mechanisms underlying the spatially differentiated structure formation via acid etching are not clearly explained. It is necessary for further detailed discussion
2. In the XRD pattern, the Miller indices corresponding to crystal planes should be indicated.
3. Comprehensive analysis including particle size, crystallite size, and BET specific surface area is necessary.
4. The XPS results reveal opposing shifts in the peaks at $2p_{1/2}$ and $2p_{3/2}$ of Ti 2p orbital, which requires explanation.
5. Discussion on the heterojunction formation within BTO should be included.
6. In SPVM, the circle marks within the figures are not distinctly clear and need further clarification.
7. Please discuss how to overcome the difficulty of crucial ratio of hydrogen to oxygen in OWS in this manuscript.
8. The manuscript lacks a coherent design or discussion regarding the Oxygen Reduction Reaction (ORR) in OWS. The introduction of cocatalysts, leading to a complex system with heterojunctions, seems to contradict the manuscript's stated significance of single component system.
9. For Figure 5f, it is suggested to present the referenced citations in an additional table for clarity.

Reviewer #2 (Remarks to the Author):

This study by Yu et al. demonstrated an intriguing strategy of modulating the Aurivillius-type layered ferroelectric photocatalyst Bi₄Ti₃O₁₂ nanosheets to produce spatially differentiated structure by a facile hydrothermal acidic treatment for enhanced photocatalytic overall water splitting. This strategy led to the efficient separation and redistribution of photo-generated charges in the Bi₄Ti₃O₁₂ nanosheets so a solar-to-hydrogen conversion efficiency of up to 0.1% under simulated solar light irradiation was achieved. The results obtained could provide some important implication for the design of high-performance layered ferroelectric photocatalytic materials. I would like to recommend its publication after considering the following comments.

- 1) It was observed that the hydrothermal acidic etching process caused the selective corrosion of lateral edges of the Bi₄Ti₃O₁₂ nanosheets instead of the basal surfaces. This means that the top basal surface (Bi₂O₂)²⁺ or perovskite layers are more stable than their internal layers in Bi₄Ti₃O₁₂ nanosheets. However, the acidic etching at room temperature in the previous study caused the selective removal of the top basal layers instead of the lateral edges for the ferroelectric Bi₃TiNbO₉ photocatalyst with very similar layered structure (Adv. Sci. 2023, 10, 2302206). What is the reason for these completely opposite results? Some theoretical calculations the etching mechanism seems necessary.
- 2) Please provide the excitation wavelengths for PL and TRPL spectroscopy characterizations. In addition, there are some confused explanations for the TRPL data. The fluorescence signal should be attributed to the radiative recombination of carriers (line 224) rather than non-radiative recombination.
- 3) The unstable photocurrent signal of the pristine Bi₄Ti₃O₁₂ (Figure S19a) was observed. On the contrary, the pristine Bi₄Ti₃O₁₂ demonstrated stable photocatalytic overall water splitting activity (Figure S20a). How to understand these opposite results?
- 4) According to the Figure 5b, the claimed 1.3% for the AQY of photocatalytic overall water splitting at 420 nm was incorrect (line 267). This value is smaller than 1%. Please check it.
- 5) Besides the enhanced charge transfer efficiency of BTO-2, some other favorable characters for BTO-2 such as large surface area, the good dispersion of cocatalysts should be considered.

Reviewer #3 (Remarks to the Author):

In this work, titled "Charge Re-distribution of a Spatially Differentiated Ferroelectric Bi₄Ti₃O₁₂ Single Crystal for Photocatalytic Overall Water Splitting", authors reported Bi₄Ti₃O₁₂ single crystal photocatalyst with spatially differentiated structure, achieving efficient charge transfer, which present a novel strategy for designing photocatalytic system. The structure-activity relationship of photocatalytic

overall water splitting of single crystal bismuth titanate was established by the changes of carrier lifetime and light absorption during the morphology evolution. The dynamic evolution of the photogenerated charge carriers was analyzed by high-resolution kelvin probe force microscopy technique. Also, spatially differentiated Bi₄Ti₃O₁₂ exhibits an excellent hydrogen and oxygen evolution for one-step-excitation OWS at a solar-to-hydrogen efficiency of 0.1% under simulated solar light, which is 212 times greater than pristine Bi₄Ti₃O₁₂. This strategy in the study shows considerable appeal. Thus, I suggest its acceptance after some revisions.

1. Recently, the reported Bi-based Bi₃TiNbO₉ ferroelectric material (Nat. Commun. 2023, 14, 7948) shows the performance of photocatalytic water splitting through internal electrostatic field effect, which is of reference significance for the current system and should be discussed.
2. The author should further explain the formation process of hollow structure at the edge of Bi₄Ti₃O₁₂ single crystal in detail, which is essential to guide the next discussion.
3. The author gives the SPVM characterization of Bi₄Ti₃O₁₂ with hollow structure at the edge. As a contrast, the corresponding SPVM characterization of pristine Bi₄Ti₃O₁₂ single crystal should be given to better understand the proposed "spatially differentiated structure".
4. The authors should count the thickness of the upper and lower layers of the hollow structure at the edge.
5. As proposed by the author, a suitable acidic environment can be conducive to edge etching. I wonder whether the etched Bi elements will once again form other structures such as bismuth oxyhalide.

Response to Reviewer

Dear Reviewers,

Thanks very much for your valuable and constructive comments on our manuscript entitled “**Charge Re-distribution of a Spatially Differentiated Ferroelectric Bi₄Ti₃O₁₂ Single Crystal for Photocatalytic Overall Water Splitting**” (Manuscript ID: NCOMMS-23-58167-T) for *Nature Communications*. We sincerely appreciate all the comments and suggestions, which are highly important for further improvements to our manuscript. We have now made detailed responses and substantial revisions accordingly. All the corrections are marked in blue in the revised manuscript, and detailed responses are given below.

Reviewer #1 (Remarks to the Author):

Comments:

Guangri Jia et al demonstrated a single-component BTO system, distinguished by a spatially differentiated structure, which exhibits exceptional efficacy in overall water splitting. The SPVM and photocatalytic performances of this system are noteworthy. However, while the manuscript underscores the significance of this advancement in the field of particulate photocatalysts, a more detailed, step-by-step clarification is required. Additionally, certain results necessitate further verification and explicit correlation with the stated significance. Therefore, after the comprehensive major revision, I would recommend this manuscript for further publication. Specific comments are as follows:

Response: We really appreciate Reviewer's valuable input and recognition of the significance of the research. We have since conducted supplementary characterizations and made extensive revisions to verify and correlate the “structure-property-performance” characteristics of this promising system, as noted in the highlighted manuscript and the responses in the following (blue) to specific questions. We sincerely hope that the revised manuscript will satisfy the stringent criteria for publication in *Nature Communications*.

1. The chemical mechanisms underlying the spatially differentiated structure formation via acid etching are not clearly explained. It is necessary for further detailed discussion
Answer 1-1: Thanks for your comment. Stability and activity have been noted to

depend on crystal facets in a wide range of crystalline photocatalysts with strong isotropic bonds because surface atomic arrangement and electronic structure vary sensitively with exposed facets. However, selective facet exposure in layered materials with anisotropic bonds seems difficult to achieve because their particle surface usually consists of only two types of facets with diverse properties, i.e. the basal {001} surface and the lateral surface parallel to the direction of layer stacking. By extensive DFT calculations (**Figure R1-1**), the stability of the $[\text{Bi}_2\text{O}_2]^{2+}$ layer structure is shown to be lower than that of the laterally exposed structure, both of which are lower than that of the exposed Ti-O in the perovskite structure of $[\text{Bi}_2\text{Ti}_3\text{O}_{10}]^{2-}$ in acidic condition. Therefore, the basal {001} surface terminated with Ti-O of $[\text{Bi}_2\text{Ti}_3\text{O}_{10}]^{2-}$ prevented the {001} plane from further etching under HCl conditions due to the above stability difference. And the lateral surface will continue to be etched and eventually form a hollow structure on the side in a stable structure.

So, to clarify this point, the following text was added to page 5, line 108-113 in the revised manuscript. “Due to the different structural stability of exposed crystal facets, the $[\text{Bi}_2\text{O}_2]^{2+}$ structure of {001} crystal facet and Bi-O bond of {010} crystal facet of BTO can be selectively preferentially etched (**Supplementary Fig. S1**). Also, the Ti-O bond of BTO {001} will prevent further {001} crystal facet from etching after etching $[\text{Bi}_2\text{O}_2]^{2+}$, ultimately leading to the formation of a hollow structure along the edge region.”

Figure R1-1. (Supplementary Figure S1) Free energy change of each step in the acid-etching reaction. R1-R3 is the M-pathway (metal dissolution). R4-R6 is the O-pathway (oxygen dissolution). According to the free energy change, the stability: $\text{Bi}_{(001)}$ ($\Delta G_{\text{max}} = 3.17 \text{ eV}$) < $\text{Bi}_{(010)}$ ($\Delta G_{\text{max}} = 12.68 \text{ eV}$) < $\text{Ti}_{(001)}$ ($\Delta G_{\text{max}} = 19.62 \text{ eV}$)

2. In the XRD pattern, the Miller indices corresponding to crystal planes should be indicated.

Answer 1-2: Thanks for your suggestions. The Miller indices corresponding to crystal

planes have been labeled in the XRD pattern (**Figure R1-2**).

Figure R1-2 (Figure 1b) XRD patterns for as-prepared BTO at different HCl treated duration.

3. Comprehensive analysis including particle size, crystallite size, and BET specific surface area is necessary.

Answer 1-3: Thanks for your suggestions. Catalytic performance is closely related to the morphology, size, and specific surface area of the catalyst. We have now analyzed particle size and crystal size (single crystal structure) by SEM and obtained the specific surface area of the material by BET (**Figure R1-3** and **Table R1-1**). By comparison, with the increase of etching depth, the particle size and crystallite size remain approximate unchanged within an error of ± 20 nm, but the specific surface area increases correspondingly, which poses the positive effect on the improvement of properties.

Figure R1-3. (Supplementary Figure S3) The particle size of basal surface of samples with and without acid etching: a) BTO, b) 0.25 h, c) 0.5 h, d) 1 hours, and e) 2 h. f) The N₂ adsorption isotherms plots.

Table R1-1. (Supplementary Table S1) The particle size (from Figure 1a), crystallite size (from Figure 1a), and BET specific surface area of BTO-2, BTO-1, BTO-0.5, BTO-0.25, and BTO.

Samples	Particle size (μm)	Crystallite size (μm)	specific surface area ($\text{m}^2 \text{g}^{-1}$)
BTO	1.09	1.09	3.6
BTO-0.25	1.11	1.11	3.6
BTO-0.5	1.10	1.10	4.2
BTO-1	1.08	1.08	4.9
BTO-2	1.10	1.10	23.3

4. The XPS results reveal opposing shifts in the peaks at 2p_{1/2} and 2p_{3/2} of Ti 2p orbital,

which requires explanation.

Answer 1-4: Thanks for pointing out the inconsistency. We reassigned the attribution of XPS peaks in **Figure R1-4 (Figure 2f)**. As we all known, the signal reflected by XPS characterization is mainly from the surface structure in the 10 nm range of the material. In the etching process, the preferentially etched structures are mainly the basal surfaces ($[\text{Bi}_2\text{O}_2]^{2+}$) and the lateral edges structure, resulting in changes in the binding energy of Bi and Ti. Due to the etching effect, including the Bi-O moiety in the $[\text{Bi}_2\text{Ti}_3\text{O}_{10}]^{2-}$ structure in perovskite structures and $[\text{Bi}_2\text{O}_2]^{2+}$, the structures exposed to the surface are predominantly Ti-O terminals. Therefore, Bi 4f peak slightly shifts downward and the binding energy peak of Ti 2p gradually shifts towards a higher value.

So, to clarify this point, the following text was added to page 7, line 155-163 in the revised manuscript. “Evidently, as shown in **Fig. 2f**, the binding energy peak of Ti 2p gradually shifts towards a higher value, which comes from the increasing proportion of the Ti-O structures exposed on the surface because of the Ti element binding more high-energy O (hydroxyls and water molecules) (**Supplementary Fig. S16a**). In contrast, the corresponding Bi 4f peak slightly shifts downward in binding energy with increasing extent of the acid etching, because the outside high binding energy Bi of $[\text{Bi}_2\text{O}_2]^{2+}$ layer and $[\text{Bi}_2\text{Ti}_3\text{O}_{10}]^{2-}$ was etched away. Although the binding energies of Bi and Ti evolve, the main structure of samples keeps unchanged during the etching process (**Fig. 2g**).”

Figure R1-4 (Figure 2f) High-resolution XPS spectra of Ti 2p.

5. Discussion on the heterojunction formation within BTO should be included.

Answer 1-5: Thanks for your comments. We have modified the term “heterojunction”

to “homojunction”. With the increase of the etching time, the etched region was deepened, which regulates the band structure (**Supplementary Fig. S17 and S18**). The band structure of the etched part is different from that of the unetched. Therefore, the structure containing different band structures but the same material forms a “homojunction”.

So, to clarify this point, the following text was added to page 7, line 166-172 in the revised manuscript. “The bandgap slightly increases with the extension of the etching, which is due to the change of the band structure because of the spatial size effect, which is mainly from ultrathin hollow edge (**Supplementary Figs. S17-S19**). With the increase of edge etching depth, the trend of band structure change is more obvious. Meanwhile, the difference in the band structures between unetched part and adjacent etched BTO will form “homojunction”, which are beneficial to the transfer of electrons to the surface of the photocatalyst.”

6. In SPVM, the circle marks within the figures are not distinctly clear and need further clarification.

Answer 1-6: Thanks for your comments. We re-labeled the circles and discuss those (**Figure R1-5**).

Figure R1-5. (Figure 3a-f) a-f, SPVM images at different wavelengths of 350 nm, 360 nm, 370 nm, 380 nm, 400 nm, and 420 nm. Blue circle (α_1), black circle (α_2) and white circle (α_3) represent the center region of BTO-2, the edge region of BTO-2, and the sapphire substrate, respectively.

7. Please discuss how to overcome the difficulty of crucial ratio of hydrogen to oxygen

in OWS in this manuscript.

Answer 1-7: Thanks for your comments. Indeed, it is very difficult to achieve stoichiometric hydrogen and oxygen production ratio of 2 to 1 from photocatalytic water splitting. The solubilities of hydrogen and oxygen in water are different. At a relatively low production rate, the ratio of hydrogen and oxygen will be seriously unbalanced, which is not due to the catalyst material itself, and is usually ignored. More importantly, the reverse reaction, side reaction, defects, and stability of the photocatalysts will affect the ratio of evolved hydrogen and oxygen. To address those issues, appropriate co-catalysts are employed to promote the photo-redox processes. Usually, co-catalysts, such as Rh-CrO_x, are deposited on the photocatalyst, to promote the forward reaction of water splitting by facilitating the occurrence of hydrogen production due to sorption and activation of H species, so as to prevent the reverse reaction and side reaction. Similarly, the cocatalyst of CoO_x facilitate oxygen generation. In addition, the photocatalyst is usually of a defect-free single-crystal structure to ensure maximum utilization of the photo-generated electrons and holes. It is also essential to have a stable catalyst material that does not cause photo-corrosion. So, to clarify this point, the following text was added to page 4, line 88-92 and 95-100, in the revised manuscript. “The specific etching does not alter the original material structure and maintains the single crystals free from grain boundaries and defects over the period of treatment, which will make sure maximum utilization of photogenerated electrons and holes without consuming either of them and no photocorrosion.” and “ To achieve stoichiometric ratio of H₂:O₂ = 2:1 and efficient performance, the suitable co-catalysts were deposited on photocatalyst in favor of the forward reaction direction to split water into hydrogen and oxygen, such as Rh-CrO_x, which promotes the occurrence of hydrogen production and prevents the reverse reaction and side reaction. And the cocatalyst of CoO_x will catalyze water to oxygen.”

8. The manuscript lacks a coherent design or discussion regarding the Oxygen Reduction Reaction (ORR) in OWS. The introduction of cocatalysts, leading to a complex system with heterojunctions, seems to contradict the manuscript's stated significance of single component system.

Answer 1-8: Thanks for your comments. Usually, we deposit co-catalysts on a photocatalyst in favor of the forward reaction to split water into hydrogen and oxygen, such as Rh-CrO_x, which promotes the occurrence of hydrogen evolution and prevents

the reverse reaction and side reactions, such as ORR. And the cocatalyst of CoO_x will catalyze water to oxygen. Also, for “single component system” in photocatalysis, this concept contrasts with complex systems, such as Type-II and Z-scheme. In the photocatalytic system presented here, the photogenerated electrons and holes are only generated by a single material BTO. There is no carrier transfer (loss) between different materials, so it is a single component system to impart high efficiency.

So, to clarify this point, the following text was added to page 4, line 95-100, in the revised manuscript. “To achieve stoichiometric ratio of $\text{H}_2:\text{O}_2 = 2:1$ and efficient performance, the suitable co-catalysts were deposited on photocatalyst in favor of the forward reaction direction to split water into hydrogen and oxygen, such as Rh-CrO_x , which promotes the occurrence of hydrogen production and prevents the reverse reaction and side reaction. And the cocatalyst of CoO_x will catalyze water to oxygen.”

9. For Figure 5f, it is suggested to present the referenced citations in an additional table for clarity.

Answer 1-9: Thanks for your suggestion. We have presented the referenced citations in an additional table (**Table R1-2**).

Table R1-2. (Supplementary Table S2) The referenced citations in Figure 5f.

Materials	H_2 ($\mu\text{mol/h}$, xenon lamp)	STH (%)	Reference number
$\text{Bi}_4\text{Ti}_3\text{O}_{12}$	378	0.1	This work
$\text{Y}_2\text{Ti}_2\text{O}_5\text{S}_2$	7.5	0.007	Ref. 9
Zr-TaON	15	0.009	Ref. 24
Organolead iodide	3.1	0.014	Ref. 25
Mg-BaTaO ₂ N	2	0.0004	Ref. 26
Ta ₃ N ₅ nanorod	11	0.014	Ref. 54
ZnIn ₂ S ₄	56.6	0.003	Ref. 55
Conjugated Polymer	10.9	0.0047	Ref. 56
SrTaO ₂ N	9.1	0.0063	Ref. 57
β -ketoamine COF	2	0.23	Ref. 58

Reviewer #2 (Remarks to the Author):

Comments:

This study by Yu et al. demonstrated an intriguing strategy of modulating the Aurivillius-type layered ferroelectric photocatalyst $\text{Bi}_4\text{Ti}_3\text{O}_{12}$ nanosheets to produce spatially differentiated structure by a facile hydrothermal acidic treatment for enhanced

photocatalytic overall water splitting. This strategy led to the efficient separation and redistribution of photo-generated charges in the $\text{Bi}_4\text{Ti}_3\text{O}_{12}$ nanosheets so a solar-to-hydrogen conversion efficiency of up to 0.1% under simulated solar light irradiation was achieved. The results obtained could provide some important implication for the design of high-performance layered ferroelectric photocatalytic materials. I would like to recommend its publication after considering the following comments.

Response: We really appreciate the kind comments, which hold immense value in shaping the revision of our work. We have since dedicated our efforts to offer further explanations and supplementary descriptions to address the concerns, as highlighted below.

1. It was observed that the hydrothermal acidic etching process caused the selective corrosion of lateral edges of the $\text{Bi}_4\text{Ti}_3\text{O}_{12}$ nanosheets instead of the basal surfaces. This means that the top basal surface $(\text{Bi}_2\text{O}_2)^{2+}$ or perovskite layers are more stable than their internal layers in $\text{Bi}_4\text{Ti}_3\text{O}_{12}$ nanosheets. However, the acidic etching at room temperature in the previous study caused the selective removal of the top basal layers instead of the lateral edges for the ferroelectric $\text{Bi}_3\text{TiNbO}_9$ photocatalyst with very similar layered structure (Adv. Sci. 2023, 10, 2302206). What is the reason for these completely opposite results? Some theoretical calculations the etching mechanism seems necessary.

Answer 2-1: Thanks for your comments. The room temperature conditions and low acid concentration used in the literature mentioned by the reviewers are different from the conditions mentioned in our experiment. The etching effect of $(\text{Bi}_2\text{O}_2)^{2+}$ structure is also mentioned in our manuscript, which is consistent with the results in the literature mentioned by the reviewer. The difference is that it is not only the {001} surface $(\text{Bi}_2\text{O}_2)^{2+}$ structure that is etched, but also the side {010} crystal face that is etched over a long period of time. In order to better explain this result, we have carried out theoretical calculation to analyze the structural stability of different crystal faces and structure. Stability and activity have been noted to depend on crystal facets in a wide range of crystalline photocatalysts with strong isotropic bonds because surface atomic arrangement and electronic structure vary sensitively with exposed facets. By extensive DFT calculations (**Figure R1-1**), the stability of the $[\text{Bi}_2\text{O}_2]^{2+}$ layer structure is shown to be lower than that of the laterally exposed structure, both of which are lower than that of the exposed Ti-O in the perovskite structure of $[\text{Bi}_2\text{Ti}_3\text{O}_{10}]^{2-}$ in acidic condition. Therefore, the basal {001} surface terminated with Ti-O of $[\text{Bi}_2\text{Ti}_3\text{O}_{10}]^{2-}$ prevented the

{001} plane from further etching under HCl conditions due to the above stability difference. And the lateral surface will continue to be etched and eventually form a hollow structure on the side in a stable structure.

So, to clarify this point, the following text was added to page 5, line 108-113 in the revised manuscript. “Due to the different structural stability of exposed crystal facets, the $[\text{Bi}_2\text{O}_2]^{2+}$ structure of {001} crystal facet and Bi-O bond of {010} crystal facet of BTO can be selectively preferentially etched (**Supplementary Fig. S1**). Also, the Ti-O bond of BTO {001} will prevent further {001} crystal facet from etching after etching $[\text{Bi}_2\text{O}_2]^{2+}$, ultimately leading to the formation of a hollow structure along the edge region.”

2. Please provide the excitation wavelengths for PL and TRPL spectroscopy characterizations. In addition, there are some confused explanations for the TRPL data. The fluorescence signal should be attributed to the radiative recombination of carriers (line 224) rather than non-radiative recombination.

Answer 2-2: Thanks for your suggestions. We have added the relevant parameters in the **Method** part of the Manuscript. We revised and discussed the TRPL results in detail. So, to clarify this point, the following text was added to page 10-11, line 229-237 in the revised manuscript. “The average lifetime is 3.64 ns for pristine BTO, which is longer than that of the etched BTO, and the average carrier lifetime reduces with the duration of HCl treatment (3.28 ns for BTO-0.25, 2.71 ns for BTO-0.5, 2.34 ns for BTO-1, and 1.69 ns for BTO-2), representing that the etched BTO enhances the kinetics of charge carrier transfer. Particularly, the decay lifetime of τ_1 of the etched BTO is longer than that of pristine BTO and the contribution of fast decay of τ_1 increases largely from 34% for BTO to 85% for BTO-2, confirming the suppression of the inter-band recombination and fast electron transfer of the etched BTO due to the homojunction between the edge and the center.”

3. The unstable photocurrent signal of the pristine $\text{Bi}_4\text{Ti}_3\text{O}_{12}$ (Figure S19a) was observed. On the contrary, the pristine $\text{Bi}_4\text{Ti}_3\text{O}_{12}$ demonstrated stable photocatalytic overall water splitting activity (Figure S20a). How to understand these opposite results?

Answer 2-3: Thanks for your comments. These “seemingly” opposite results can be attributed to the different conditions of the two experiments. In the photocurrent experiment, due to the instantaneous irradiation of the light source on the electrode

surface, the photoexcited carrier transition will lead to the instantaneous enhancement of the photocurrent. Then the photogenerated carrier recombination and transfer will reach a balance, and the photocurrent will gradually attain a stable state, which is related to the transfer rate of the photogenerated carriers. Because the carrier transfer of BTO is slow, the photocurrent will decline slowly over a short period of time, which is not caused by the insufficient stability of the material itself. With the extension of time, the photocurrent will maintain at a constant value.

4. According to the Figure 5b, the claimed 1.3% for the AQY of photocatalytic overall water splitting at 420 nm was incorrect (line 267). This value is smaller than 1%. Please check it.

Answer 2-4: Thanks for your comments. We have double-checked and revised it.

So, to clarify this point, the following text was added to page 13, line 276-277 in the revised manuscript. “The AQE value of OWS reaches 5.0% at 360 nm, 2.1% at 380 nm, and 1.3% at 400 nm, respectively.”

5. Besides the enhanced charge transfer efficiency of BTO-2, some other favorable characters for BTO-2 such as large surface area, the good dispersion of cocatalysts should be considered.

Answer 2-5: Thank you very much for the reviewer’s suggestion. We have analyzed particle size, crystal size and specific surface area (**Figure R1-3** and **Table R1-1**).

So, to clarify this point, the following text was added to page 5, line 116-119 and page 13, line 287-289 in the revised manuscript. “However, the particle size or crystallite size of samples have no significant change but the obvious change of specific surface area before and after acid etching (**Supplementary Fig. S3** and **Table. S1**).” and “Also, the larger specific surface area of BTO-2 increases the dispersion of the cocatalyst, which is favorable for the improvement of photocatalytic performance.”

Reviewer #3 (Remarks to the Author):

In this work, titled “Charge Re-distribution of a Spatially Differentiated Ferroelectric $\text{Bi}_4\text{Ti}_3\text{O}_{12}$ Single Crystal for Photocatalytic Overall Water Splitting”, authors reported $\text{Bi}_4\text{Ti}_3\text{O}_{12}$ single crystal photocatalyst with spatially differentiated structure, achieving efficient charge transfer, which present a novel strategy for designing photocatalytic

system. The structure-activity relationship of photocatalytic overall water splitting of single crystal bismuth titanate was established by the changes of carrier lifetime and light absorption during the morphology evolution. The dynamic evolution of the photogenerated charge carriers was analyzed by high-resolution kelvin probe force microscopy technique. Also, spatially differentiated $\text{Bi}_4\text{Ti}_3\text{O}_{12}$ exhibits an excellent hydrogen and oxygen evolution for one-step-excitation OWS at a solar-to-hydrogen efficiency of 0.1% under simulated solar light, which is 212 times greater than pristine $\text{Bi}_4\text{Ti}_3\text{O}_{12}$. This strategy in the study shows considerable appeal. Thus, I suggest its acceptance after some revisions.

Response: Thanks very much for the kind comments. The feedback has been incredibly helpful in guiding us towards a more thorough and effective revision. We have made substantial changes to the manuscript to address the concerns and incorporate suggestions.

1. Recently, the reported Bi-based $\text{Bi}_3\text{TiNbO}_9$ ferroelectric material (Nat. Commun. 2023, 14, 7948) shows the performance of photocatalytic water splitting through internal electrostatic field effect, which is of reference significance for the current system and should be discussed.

Answer 3-1: Many thanks for the reviewer's suggestions. We have referred to the literature mentioned above, and discussed it in the Introduction and Reference **Ref. 39**. The corresponding content is “Such a sandwiched structure may facilitate the separation of photogenerated electron-hole pairs by means of the internal electric field formed between the interlayers, especially the introduction of external factors to enhance this effect.” in the **Introduction**.

So, to clarify this point, the following text was added to page 3, line 65-67 in the revised manuscript. “Such a sandwiched structure may facilitate the separation of photogenerated electron-hole pairs by means of the internal electric field formed between the interlayers, especially the introduction of external factors to enhance this effect.³⁶⁻³⁹”

2. The author should further explain the formation process of hollow structure at the edge of $\text{Bi}_4\text{Ti}_3\text{O}_{12}$ single crystal in detail, which is essential to guide the next discussion.

Answer 3-2: Thanks for your constructive comments. Different crystal facets stability

are associated with anisotropic bonds in layered materials, which leads to two types of facets with diverse properties for BTO, i.e. the basal {001} surface and the lateral surface parallel to the direction of layer stacking. By extensive DFT calculations (**Figure R1-1**), the stability of the $[\text{Bi}_2\text{O}_2]^{2+}$ layer structure is shown to be lower than that of the laterally exposed structure, both of which are lower than that of the exposed Ti-O in the perovskite structure of $[\text{Bi}_2\text{Ti}_3\text{O}_{10}]^{2-}$ in acidic condition. Therefore, the basal {001} surface terminated with Ti-O of $[\text{Bi}_2\text{Ti}_3\text{O}_{10}]^{2-}$ prevented the {001} plane from further etching under HCl conditions due to the above stability difference. And the lateral surface will continue to be etched and eventually form a hollow structure on the side in a stable structure.

So, to clarify this point, the following text was added to page 5, line 108-113 in the revised manuscript. “Due to the different structural stability of exposed crystal facets, the $[\text{Bi}_2\text{O}_2]^{2+}$ structure of {001} crystal facet and Bi-O bond of {010} crystal facet of BTO can be selectively preferentially etched (**Supplementary Fig. S1**). Also, the Ti-O bond of BTO {001} will prevent further {001} crystal facet from etching after etching $[\text{Bi}_2\text{O}_2]^{2+}$, ultimately leading to the formation of a hollow structure along the edge region.”

3. The author gives the SPVM characterization of $\text{Bi}_4\text{Ti}_3\text{O}_{12}$ with hollow structure at the edge. As a contrast, the corresponding SPVM characterization of pristine $\text{Bi}_4\text{Ti}_3\text{O}_{12}$ single crystal should be given to better understand the proposed “spatially differentiated structure”.

Answer 3-3: Many thanks to the helpful comments and suggestion. Accordingly, we have added the corresponding SPVM characterization of pristine $\text{Bi}_4\text{Ti}_3\text{O}_{12}$ (**Figure R3-1**).

Figure R3-1. (Figure S20) a) AFM image of BTO. b-g) SPVM images of BTO at different wavelengths of 350 nm, 360 nm, 370 nm, 380 nm, 400 nm, and 420 nm. h) SRSPS of BTO nanoparticle as a function of photoexcited wavelength under illumination.

4. The authors should count the thickness of the upper and lower layers of the hollow structure at the edge.

Answer 3-4: Many thanks to your suggestion. For the thickness of the upper and lower layers of the hollow structure at the edge, we conducted data analysis on the lateral edges (**Supplementary Fig. S12**). The results show the thickness of upper and lower layers are both ~ 20 nm.

5. As proposed by the author, a suitable acidic environment can be conducive to edge etching. I wonder whether the etched Bi elements will once again form other structures such as bismuth oxyhalide.

Answer 3-5: Many thanks to your comments. We proved that no other phases, e.g. bismuth oxyhalide structure, appeared during our experimental synthesis through XRD characterization (**Figure 1b**). Moreover, the Cl element was not detected in the surface (**Supplementary Fig. S16**). Therefore, in this synthesis process, the etched Bi element did not nucleate again to form a second phase material. As a result, as the reviewer mentioned, a certain acid system is not only the etching effect, but also prevents the nucleation and growth of the second phase material to a certain extent.

REVIEWERS' COMMENTS

Reviewer #2 (Remarks to the Author):

The authors conducted the satisfactory revisions to the manuscript. I would like to recommend the publication of this version in NC.

Reviewer #3 (Remarks to the Author):

The authors have answered the question we raised. We recommend to accept this manuscript.